# AlignedGen: Aligning Style Across Generated Images

**Jiexuan Zhang**[1*], **Yiheng Du**[1*], **Qian Wang**[1], **Weiqi Li**[1], **Yu Gu**[1], **Jian Zhang**[1,2✉]

[1]School of Electronic and Computer Engineering, Peking University
[2]Guangdong Provincial Key Laboratory of Ultra High Definition Immersive Media Technology,
Shenzhen Graduate School, Peking University

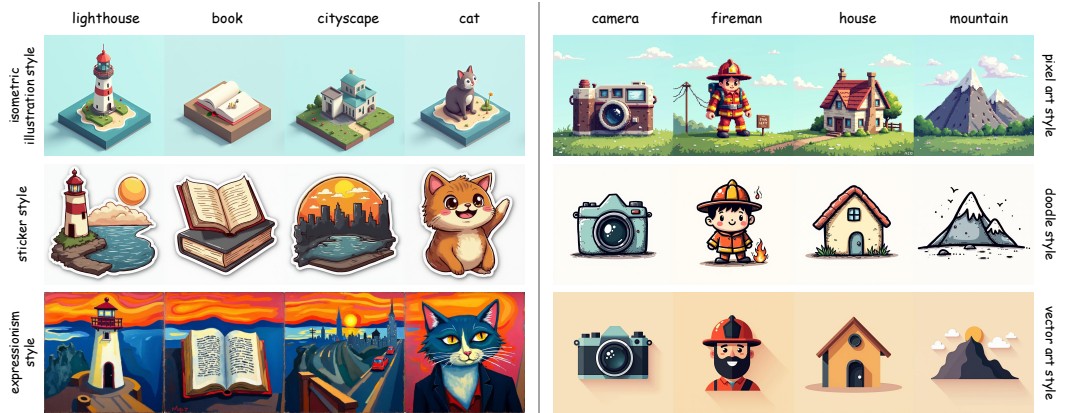

Figure 1: **Generation results of AlignedGen.** As a training-free method, AlignedGen generates images with varying content while ensuring high style consistency.

## Abstract

Diffusion-based generative models struggle to maintain high style consistency across generated images via text description. Although several style-aligned image generation methods have been proposed to address this issue, they exhibit suboptimal performance and are primarily built upon the U-Net architecture, limiting their compatibility with DiT diffusion models like Flux that has emerged as a predominant model in the field of image generation. To address these limitations, we propose *AlignedGen*, a novel training-free style-aligned image generation method for DiT models to significantly enhance style consistency across generated images. Specifically, AlignedGen incorporates two key components to achieve this: Shifted Position Embedding (ShiftPE) and Advanced Attention Sharing (AAS). ShiftPE alleviates the text controllability degradation observed in prior methods when applied to DiT models through its non-overlapping position indices design, while AAS comprises three specialized techniques to unleash the full potential of DiT for style-aligned generation. Furthermore, to broaden the applicability of our method, we present an efficient query, key, and value feature extraction algorithm, enabling our method to seamlessly incorporate external images as style references. Extensive experimental results validate that our method effectively enhances style consistency across generated images while maintaining favorable text controllability. Code: https://github.com/Jiexuanz/AlignedGen.

*Equal Contribution. ✉: Corresponding author, zhangjian.sz@pku.edu.cn. This work was supported in part by National Natural Science Foundation of China (No. 62372016) and by Guangdong Provincial Key Laboratory of Ultra High Definition Immersive Media Technology (No. 2024B1212010006).

39th Conference on Neural Information Processing Systems (NeurIPS 2025).

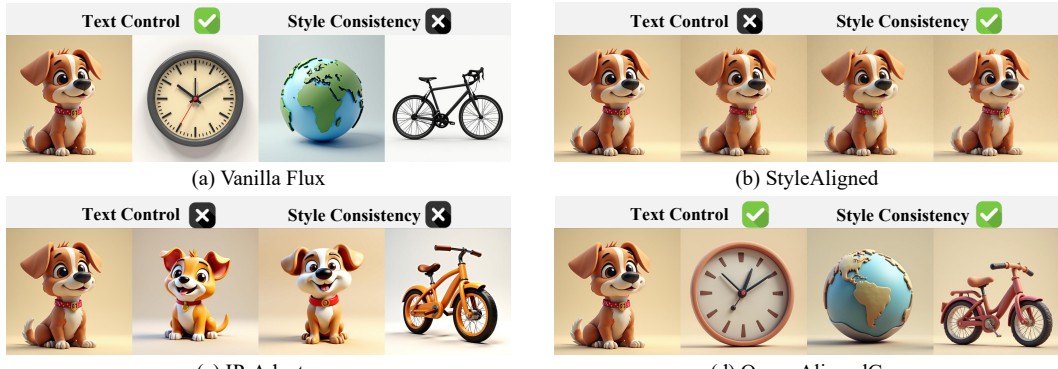

Figure 2: **Generated results using different methods on Flux with the prompt "{Dog, Clock, Globe, Bicycle} in 3D realism style".** The Vanilla Flux produces outputs with style discontinuities. The application of StyleAligned [16] on Flux leads to severe text controllability degradation. The results generated by IP-Adapter [50] perform suboptimally in text control and style consistency. Our method effectively enhances style consistency while maintaining alignment with the text prompt.

## 1   Introduction

Large-scale text-to-image diffusion models [17, 36, 33, 8, 1] empower users to create imaginative scenes from text. Early diffusion models commonly adopted the U-Net [37] as their backbone. Recently, diffusion models based on DiT, such as Flux [1], have emerged as mainstream architectures and are now the most widely used frameworks in this domain. However, all diffusion models struggle to maintain high style consistency across a series of generated images via prompts, as shown in Figure 2 (a). Such style consistency is crucial for many applications: from illustrating books and stories, through designing virtual assets, to creating graphic novels and synthetic data.

Common approaches [10, 39, 9, 41] for maintaining style consistency across generated images primarily rely on fine-tuning of the diffusion model over one image or a set of images that share the same style. However, these fine-tuning-based methods suffer from significant limitations: collecting new data that share the same style and retraining diffusion model for each target style substantially increase temporal and computational costs. Alternative approaches [26, 34, 11, 49, 45, 19] focus on constructing a style encoder combined with re-trained image-conditioned diffusion models. However, creating high-quality training datasets remains challenging due to defining "style" is challenging, leading to suboptimal performance of these methods, Figure 2 (c) illustrates this.

StyleAligned [16], a training-free approach for style-aligned image generation, has emerged to address these limitations. While promising, its reliance on the aging SDXL architecture inherently limits its output quality, often producing artifacts. Most critically, its core strategy is architecturally incompatible with DiT – the superior generative architecture for diffusion models. Designed explicitly for U-Net's self-attention, it fails when applied to DiT, resulting in loss of text control and severe content collapse, as shown in Figure 2 (b). This reveals an unaddressed gap – one that our work addresses: how to achieve robust, training-free style-aligned image generation for DiT models.

To address the limitations of existing methods, we propose *AlignedGen*, a novel framework to enhance style consistency across images generated by DiT-based diffusion model like Flux. First, we conduct a critical investigation into a fundamental problem: the failure of attention sharing mechanisms in the DiT. We identify the root cause as an improper setup of Rotary Position Embedding (RoPE), which creates conflicting positional signals. To resolve this, we propose **Shift**ed **P**osition **E**mbedding (**ShiftPE**), a novel and elegant solution that assigns non-overlapping positional spaces to each image. This not only fixes generation collapse and restores text controllability but also offers a crucial insight for all future attention sharing works on DiT. Building upon this breakthrough, we further develop **A**dvanced **A**ttention **S**haring (**AAS**), a suite of three specialized techniques - Selective Shared Attention, Controllable Style Consistency Via Key Scaling, and Layer-Selective Application. These techniques are specifically designed for the DiT architecture, further unleashing its generative potential and leading to superior output quality. Finally, to extend our method's utility, we introduce an approximate query, key, value extraction algorithm, allowing it to condition on external images

as style references and expanding its application scope. Extensive experiments demonstrate that our method significantly outperforming prior work in both style consistency and faithfulness to text prompts. In summary, our contributions are as follows:

❏ (1) We present AlignedGen, the pioneering training-free style-aligned image generation framework designed for DiT. AlignedGen operates without any fine-tuning or extra modules, enabling the efficient utilization of large DiT models with billions of parameters.

❏ (2) We discover that conflicting position embeddings are the root cause of attention sharing failure in DiT. We introduce Shifted Position Embedding (ShiftPE), a novel and effective solution that enables viable attention sharing in DiT architecture for the first time.

❏ (3) We propose Advanced Attention Sharing (AAS), a set of three specialized techniques engineered to unleash the full potential of DiT for style-aligned generation, markedly improving style fidelity and image coherence.

❏ (4) We devise an algorithm to extract approximate query, key, and value features from any external image. This allows our framework to support user-provided images as style references without retraining, significantly broadening its applicability.

## 2  Related Work

### 2.1  Attention Control in Diffusion Models

Recent years, diffusion-based generative models [17, 30, 40, 36, 33, 8, 1] have made significant advancements in image generation. This progress has been accompanied by a series of studies focusing on attention mechanisms [15, 43, 4, 28, 25, 29, 22, 48, 23, 47, 14]. Hertz et al. [15] examined the pivotal role of cross-attention in controlling image layout and content, introducing a generation control method guided by attention maps. Plug-and-Play [43] leverages spatial features and self-attention maps from the original image to guide text-conditioned image-to-image translation, effectively preserving the original image's spatial layout. Notably, these studies all rely on the design of self-attention and cross-attention mechanisms. With the advent of Transformer-based diffusion models [32, 8, 1], self-attention and cross-attention are integrated into a single attention mechanism, presenting adaptability challenges for previous works based on separate self-attention and cross-attention designs.

### 2.2  Style Image Generation

Style image generation aims to produce images that embody specific artistic styles. Early studies in this domain concentrated on style transfer techniques [51, 12, 3, 6, 20, 24, 27, 52], which enable the transfer of the content from one image to the style of another. Generative models[13, 17, 33, 1] enable widespread adoption for style image generation. Tuning-based methods [18, 21, 39, 10, 9, 41] optimize model parameters or text embeddings to capture style of images, but suffer from training overhead. Alternatively, dataset-driven approaches [34, 11, 49, 26, 19] train diffusion models on curated datasets, yet face challenges in constructing high-quality style datasets. More pertinent to our approach is StyleAligned [16], which aligns the styles of generated images by sharing information between self-attention layers without any form of optimization or fine-tuning. In contrast to other style image generation methods, it places greater emphasis on addressing the issue of inconsistent understanding of style descriptors by generative models during image generation.

## 3  Method

### 3.1  Preliminary

**Diffusion Transformer (DiT).**  DiT [32] models enhance generative performance through Transformer-based architectures, which process latent representations such as image tokens $\mathbf{X} \in \mathbb{R}^{N \times d}$ and text tokens $\mathbf{C} \in \mathbb{R}^{M \times d}$, where d is the embedding dimension and $N/M$ denote sequence lengths. In addition, these models typically encode positional information into tokens through Rotary Position Embedding (RoPE) [42], thereby embedding positional awareness into each token to achieve enhanced generation quality.

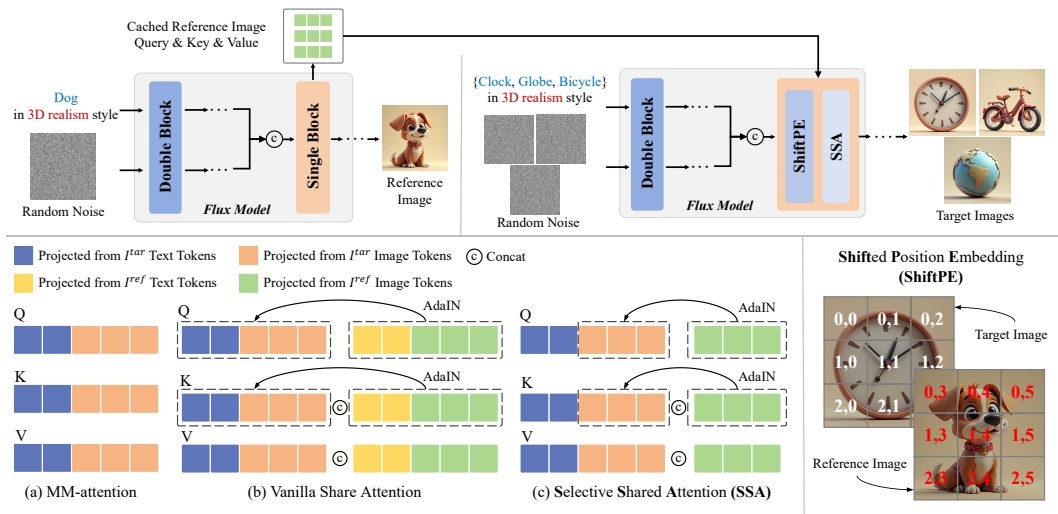

Figure 3: **Overview of the pipeline.** ShiftPE and SSA are integrated into specific layers of Flux, replacing the MM-Attention.

**Multi-Modal Attention(MM-Attention).** Some DiT models (e.g. Flux [1]) employ multi-modal attention to replace the self-attention and cross-attention mechanisms commonly used in traditional diffusion models. Specifically, image tokens $\mathbf{X}$ and text tokens $\mathbf{C}$ are projected through a set of projection matrices to obtain the corresponding representations of query $\mathbf{Q}_{\{img,txt\}}$, key $\mathbf{K}_{\{img,txt\}}$ and value $\mathbf{V}_{\{img,txt\}}$. This process is formulated as follows:

$$
\begin{aligned}
\{\mathbf{Q},\mathbf{K},\mathbf{V}\}_{img} &= \mathbf{W}^{\{Q,K,V\}}_{img}\mathbf{X}, \ \{\mathbf{Q},\mathbf{K},\mathbf{V}\}_{img} \in \mathbb{R}^{N \times d_k} \\
\{\mathbf{Q},\mathbf{K},\mathbf{V}\}_{txt} &= \mathbf{W}^{\{Q,K,V\}}_{txt}\mathbf{C}, \ \{\mathbf{Q},\mathbf{K},\mathbf{V}\}_{txt} \in \mathbb{R}^{M \times d_k}
\end{aligned}
\tag{1}
$$

where $\mathbf{W}^{\{Q,K,V\}}_{\{img, txt\}}$ denote the projection matrices for the image and text tokens, $d_k$ is the embedding dimension in attention. The $\mathbf{Q}_{\{img,txt\}}$, $\mathbf{K}_{\{img,txt\}}$ and $\mathbf{V}_{\{img,txt\}}$ are concatenated along the sequence length dimension to obtain the complete $\mathbf{Q} \in \mathbb{R}^{(M+N) \times d_k}$, $\mathbf{K} \in \mathbb{R}^{(M+N) \times d_k}$, and $\mathbf{V} \in \mathbb{R}^{(M+N) \times d_k}$:

$$
\mathbf{Q} = \text{Concat}(\mathbf{Q}_{txt}, \mathbf{Q}_{img}), \ \mathbf{K} = \text{Concat}(\mathbf{K}_{txt}, \mathbf{K}_{img}), \ \mathbf{V} = \text{Concat}(\mathbf{V}_{txt}, \mathbf{V}_{img}).
\tag{2}
$$

After that, the final output is computed using the following formula:

$$
\text{Attention}(\mathbf{Q}, \mathbf{K}, \mathbf{V}) = \text{Softmax}\left(\frac{\mathbf{Q}\mathbf{K}^{\text{T}}}{\sqrt{d_k}}\right)\mathbf{V}.
\tag{3}
$$

## 3.2 Task Formulation

In this paper, our objective is to generate a set of images $\mathcal{I} = \{I_1, I_2, ..., I_N\}$ that align with an input text prompts set $\mathcal{T} = \{T_1, T_2, ..., T_N\}$, while maintaining consistent style across all generated images. Notably, each prompt in the input text prompts set $\mathcal{T}$ incorporates a shared style descriptor. To facilitate the achievement of this objective, we designate one image in the image set $\mathcal{I}$ as the reference image $I^{ref}$, while the remaining images are defined as target images $I^{tar}$. The style of all target images $I^{tar}$ are enforced to approximate the style of the reference image $I^{ref}$. Furthermore, the query, key, and value representations of the reference image $I^{ref}$ throughout the generation process are denoted as $\mathbf{Q}^{ref}$, $\mathbf{K}^{ref}$, and $\mathbf{V}^{ref}$, respectively. The query, key, and value representations of the target images $I^{tar}$ are denoted as $\mathbf{Q}^{tar}$, $\mathbf{K}^{tar}$, and $\mathbf{V}^{tar}$.

## 3.3 Vanilla Attention Sharing

Existing style-aligned image generation technique[16] based on U-Net [37] architecture and self-attention can be directly applied to DiT diffusion models. For clarity, we refer to this approach as

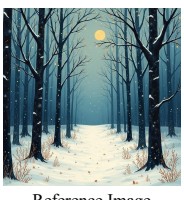 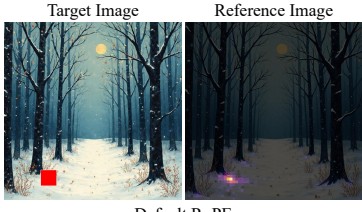 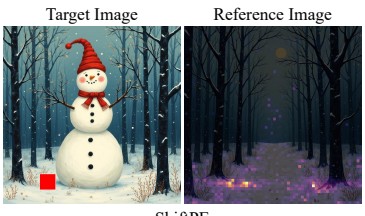

Target Image   Reference Image     Target Image   Reference Image

Reference Image              Default RoPE                    ShiftPE

Figure 4: **Visualization of attention map.** We visualize the attention map corresponding to the query feature indicated by the red box on the generated image $I^{tar}$. When using RoPE, $I^{tar}$ over-attends to spatially aligned regions in $I^{ref}$, leading to high similarity between generated and reference images. ShiftPE introduces shifted positional indices for $I^{ref}$, enabling more appropriate attention to semantically corresponding regions (e.g., attention weights spread around the snowy area periphery).

**Vanilla Attention Sharing**. Specifically, for the query, the vanilla attention sharing aligns $\mathbf{Q}^{tar}$ with $\mathbf{Q}^{ref}$ through Adaptive Instance Normalization (AdaIN)[20] to form the final query $\mathbf{Q}^F$. For the key, $\mathbf{K}^{tar}$ is aligned with $\mathbf{K}^{ref}$ through AdaIN to produce the transformed key $\hat{\mathbf{K}}^{tar}$. After which, $\hat{\mathbf{K}}^{tar}$ is concatenated with $\mathbf{K}^{ref}$ along the sequence length dimension to form the final key $\mathbf{K}^F$. For the value, vanilla attention sharing directly concatenates $\mathbf{V}^{tar}$ and $\mathbf{V}^{ref}$ along the sequence length dimension to construct the ultimate value $\mathbf{V}^F$. This process is formulated as follows:

$$
\begin{aligned}
&\text{AdaIN}(x, y) = \sigma(y)\left(\frac{x - \mu(x)}{\sigma(x)}\right) + \mu(y), \\
&\mathbf{Q}^F = \text{AdaIN}(\mathbf{Q}^{tar}, \mathbf{Q}^{ref}) \in \mathbb{R}^{(M+N)\times d_k}, \\
&\mathbf{K}^F = \text{Concat}(\hat{\mathbf{K}}^{tar}, \mathbf{K}^{ref}) \in \mathbb{R}^{2(M+N)\times d_k}, \text{ where } \hat{\mathbf{K}}^{tar} = \text{AdaIN}(\mathbf{K}^{tar}, \mathbf{K}^{ref}), \\
&\mathbf{V}^F = \text{Concat}(\mathbf{V}^{tar}, \mathbf{V}^{ref}) \in \mathbb{R}^{2(M+N)\times d_k},
\end{aligned}
\tag{4}
$$

with $\mu(\cdot)$ and $\sigma(\cdot)$ representing the mean and standard deviation. In the target image $I^{tar}$ generation process, the query, key, and value in MM-Attention are replaced by $\mathbf{Q}^F$, $\mathbf{K}^F$, and $\mathbf{V}^F$, respectively, with the results computed according to Equation 3.

### 3.4   The limitation of Vanilla Attention Sharing in DiT

Applying Vanilla Attention Sharing to DiT, while seemingly a straightforward strategy, results in a notable collapse in performance. As illustrated in Figure 2 (b), this direct approach severely degrades text controllability and causes prominent content leakage from the reference image, leading to a catastrophic failure mode where all generated images become mere copies of the reference. This raises a critical question: **Why does Vanilla Attention Sharing fail so markedly within the DiT?**

To answer this, we conduct a systematic investigation into the interplay between attention sharing mechanism and the DiT. Our investigation reveals that the root cause is not a flaw in the attention sharing mechanism per se, but a subtle yet critical issue we term **Positional Collision**. While RoPE provides unique spatial coordinates $(i, j)$ for each token within a single image's $h \times w$ latent grid, the vanilla sharing scheme inadvertently assigns identical positional embeddings to tokens at the same coordinates $(i, j)$ in both the reference and target images. This collision induces a erroneous bias, forcing tokens to disproportionately attend to their spatial counterparts across images, irrespective of content or textual guidance, as illustrated in Figure 4 (RoPE). To resolve this, we introduce a simple yet effective solution derived from this insight: **Shifted P**osition **E**mbedding (**ShiftPE**). The core idea is to ensure the reference and target images occupy distinct, non-overlapping coordinate spaces. We achieve this by virtually placing the reference image "next to" the target images, remapping its positional indices from $(i, j)$ to $(i, j + w)$, a visual depiction is provided in Figure 3 (ShiftPE). As shown in Figure 4 (ShiftPE), ShiftPE resolves the positional collision, decoupling the spatial representations and thereby eliminating the content leakage. This frees the attention mechanism to focus on regions of high semantic and stylistic relevance, rather than being constrained by rigid spatial correspondence.

### 3.5   Advanced Attention Sharing

While ShiftPE effectively resolves the problem of positional collision in Vanilla Attention Sharing, unlocking state-of-the-art performance demands a solution that works well with the specifics of

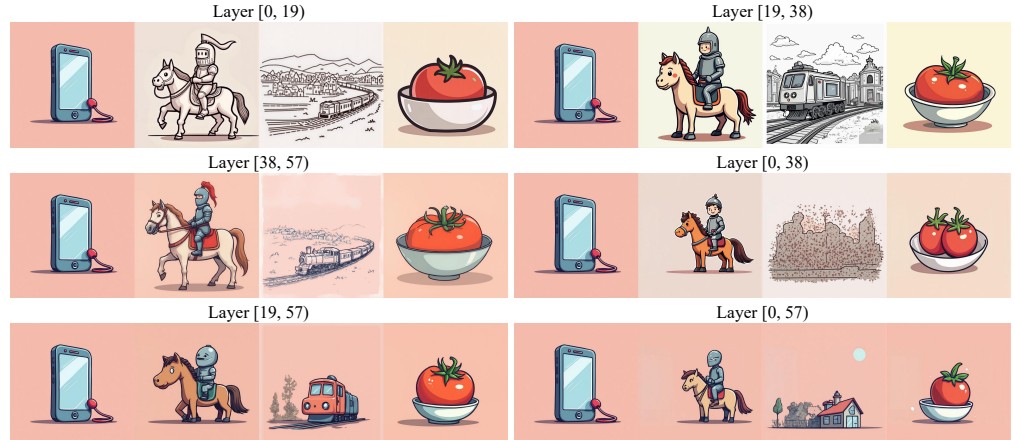

Layer [0, 19]     Layer [19, 38]

Layer [38, 57]     Layer [0, 38]

Layer [19, 57]     Layer [0, 57]

A {phone , knight on a horse, train passing a village, tomato in a bowl} in cartoon drawing style.

Figure 5: Visualization of results generated by applying our method in different layers of Flux.

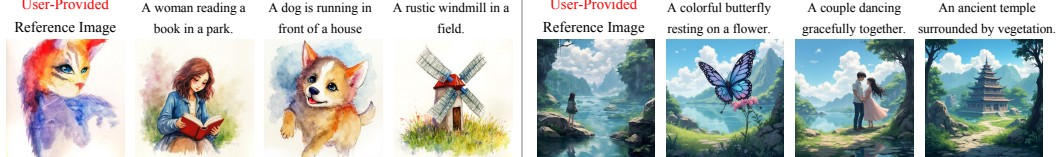

User-Provided Reference Image | A woman reading a book in a park. | A dog is running in front of a house. | A rustic windmill in a field. | User-Provided Reference Image | A colorful butterfly resting on a flower. | A couple dancing gracefully together. | An ancient temple surrounded by vegetation.

Figure 6: **Generalization from Arbitrary Style Reference.** Our method successfully generate diverse images conditioned on external, user-provided style references.

the DiT architecture. The DiT components are not inherently designed for the attention sharing. Therefore, we introduce a series of advanced modifications that enable smooth, effective cooperation with DiT's structure, significantly enhancing style consistency and generation quality.

**Selective Shared Attention.** Vanilla Attention Sharing shares the entire key and value, but it is noteworthy that in MM-Attention, the key and value are derived from both image and text tokens. This implies that full attention sharing of key and value not only propagates information from reference image but also inadvertently shares information from its corresponding prompt, which is apparently unreasonable. To address this, we propose an intuitive yet effective solution: selectively sharing key and value derived from image tokens, shown in Figure 3 This strategy not only effectively enhances the alignment between target images and their corresponding prompts but also reduces computational overhead. Finally, the $\mathbf{Q}^F$, $\mathbf{K}^F$ and $\mathbf{V}^F$ can be expressed as follow:

$$\mathbf{Q}^F = \text{Concat}(\mathbf{Q}_{txt}^{tar}, \ \hat{\mathbf{Q}}_{img}^{tar}) \in \mathbb{R}^{(M+N) \times d_k}, \ \text{where} \ \hat{\mathbf{Q}}_{img}^{tar} = \text{AdaIN}(\mathbf{Q}_{img}^{tar}, \mathbf{Q}_{img}^{ref}),$$

$$\mathbf{K}^F = \text{Concat}(\mathbf{K}_{txt}^{tar}, \ \hat{\mathbf{K}}_{img}^{tar}, \mathbf{K}_{img}^{ref}) \in \mathbb{R}^{(M+2N) \times d_k}, \text{where} \ \hat{\mathbf{K}}_{img}^{tar} = \text{AdaIN}(\mathbf{K}_{img}^{tar}, \mathbf{K}_{img}^{ref}), \quad (5)$$

$$\mathbf{V}^F = \text{Concat}(\mathbf{V}_{txt}^{tar}, \ \mathbf{V}_{img}^{tar}, \ \mathbf{V}_{img}^{ref}) \in \mathbb{R}^{(M+2N) \times d_k}.$$

**Controllable Style Consistency Via Key Scaling.** In practical applications, controlling the strength of style consistency between generated images is often required. To achieve flexible control over style consistency, we introduce a scaling factor $\lambda$. Specifically, we scale $\mathbf{K}_{img}^{ref}$ by the scaling factor $\lambda$ and concatenate it with $\mathbf{K}_{txt}^{tar}$ and $\hat{\mathbf{K}}_{img}^{tar}$ to form the modified $\mathbf{K}^F$:

$$\mathbf{K}^F = \text{Concat}(\mathbf{K}_{txt}^{tar}, \hat{\mathbf{K}}_{img}^{tar}, \lambda \cdot \mathbf{K}_{img}^{ref}). \quad (6)$$

This design allows us to adjust the attention weights by varying $\lambda$. As $\lambda$ increases, the consistency of style between the generated images becomes stronger.

**Layer-Selective Application.** The experimental results show that applying our method across all MM-Attention modules can lead to unnatural generations and textual inconsistencies in the output.

| Metric | Ours | StyleAligned [16] | IP-Adapter [50] | B-Lora [9] | StyleShot [11] | CSGO [49] |
|--------|------|-------------------|-----------------|------------|----------------|-----------|
| $S_{text} \uparrow$ | **0.282** | 0.277 | 0.278 | 0.259 | 0.276 | 0.280 |
| $S_{sty} \uparrow$ | **0.740** | 0.728 | 0.737 | 0.625 | 0.693 | 0.686 |
| $S_{dino} \uparrow$ | **0.554** | 0.512 | 0.522 | 0.271 | 0.511 | 0.450 |

Table 1: **Quantitative comparison with other methods.** We evaluate the generated image sets in terms of text alignment ($S_{text}$), style consistency ($S_{sty}$ and $S_{dino}$). The symbols $\uparrow$ indicate higher values are better. Best result is marked in **bold**, and the second-best results are marked in underline.

Inspired by prior studies [43, 7, 45, 9, 34], different layers in diffusion models capture and process distinct information. Following a similar strategy, we achieve improved performance by applying our method only within a specific subset of layers $\phi$. Specifically, setting $\phi$ to [19, 57) (the Single Blocks of Flux) yields optimal results. As shown in Figure 5, this approach not only enhances the naturalness of generated images but also maintains consistency with prompts and preserves stylistic coherence across image collections. Further experimental details can be found in Section 4.3.

## 3.6 Seamlessly Adapt to Any Style Reference

A core principle of our method is the injection of style information via attention sharing, which is predicated on access to the **Q**, **K**, and **V** features from the style reference image. While these features are directly available if the reference is synthesized by the diffusion model itself, this assumption severely limits the method's applicability in real-world cases where users provide their own images. To bridge this critical gap, we propose a novel and simple feature extraction pipeline for arbitrary external images. This pipeline achieves this by directly leveraging the forward diffusion process. For any given timestep t, we first add a noise to the reference image's latent representation. This procedure yields a noisy latent, which emulates the intermediate state during the standard generation process. By feeding this noisy latent into the diffusion model, we can query its intermediate layers to obtain the **Q**, **K**, and **V** features. We showcase qualitative results in Figure 6 and provide pseudocode in the Appendix A.

# 4 Experiment

## 4.1 Implementation Detail

**Settings.** We apply our method on the Flux.1-dev [1] by replacing the MM-Attention in all Single blocks of Flux.1-dev with ShiftPE and AAS. For inference, we use the vanilla Rectified Flow sampler with 30 sampling steps and set the classifier-free guidance scale to 3.5. Our evaluation dataset consists of 100 prompt sets from StyleAligned[16]. These prompts cover a wide range of generation targets and a diverse set of style descriptions.

**Metrics.** **(1) Text Alignment** ($S_{text}$)**.** To evaluate how well the images match the prompts, we calculate the cosine similarity using CLIP [35] between the images and prompts. **(2) Style Consistency** ($S_{sty}$)**.** To evaluate the style consistency across generated images, we calculate the pairwise average cosine similarity of the CLIP embeddings within the set of generated images. **(3) DINO** ($S_{dino}$)**.** Following [44, 39, 16], we assess style consistency by computing the pairwise average cosine similarity of DINO [5, 31] embeddings in the generated image set. This choice is motivated by the fact that CLIP, trained with category labels, often rates images with similar content but different styles as highly similar, DINO is trained in a self-supervised manner, better differentiates styles.

## 4.2 Comparison

**Competing Methods.** We compare the proposed AlignedGen against five style-aligned image generation methods: StyleAligned [16], IP-Adapter [50] (we adopt the IP-Adapter architecture and weights designed by the Instant-X team [2] for Flux.1-dev, rather than utilizing the IP-Adapter architecture based on SDXL or SD1.5), B-Lora [9], StyleShot [11], CSGO [49].

**Quantitative Comparison.** Table 1 presents the quantitative comparison results between our method and state-of-the-art style-aligned image generation approaches on the test set. Our method

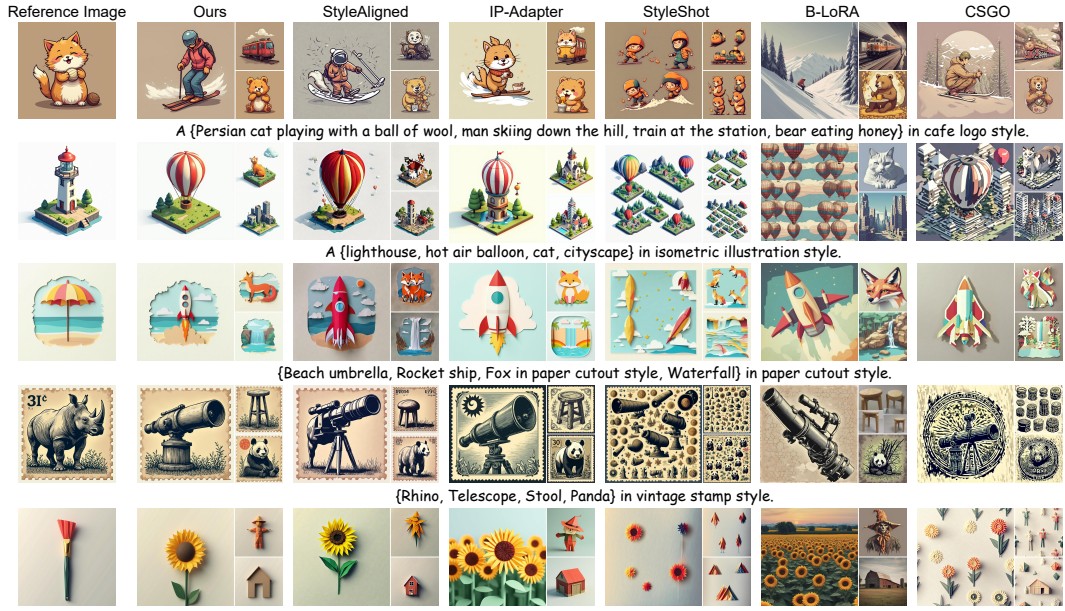

Figure 7: **Qualitative comparison between our method and other approaches.** Our method generates images with superior style consistency and accurate alignment with text prompts.

| Ours | StyleAligned [16] | IP-Adapter [50] | B-Lora [9] | StyleShot [11] | CSGO [49] |
|------|-------------------|-----------------|------------|----------------|-----------|
| **1.20** | 3.66 | 2.04 | 4.76 | 4.09 | 5.25 |

Table 2: **User Study.** The results represent the average ranking outcomes for each method (lower is better). Our approach significantly outperformed other comparative methods in the user study.

achieves the highest scores in terms of $S_{text}$, $S_{sty}$, and $S_{dino}$ metrics, demonstrating that it establishes the optimal balance between text-image alignment and style consistency among generated images.

**Qualitative Comparison.**   Figure 7 presents a qualitative comparison between our method and other approaches. IP-Adapter struggles to capture style effectively, failing to maintain consistent style in generated images and introducing content leakage issues. B-Lora, due to training its LoRA with only a single image, results in highly challenging training, consequently leading to generated images that struggle to capture the style of the reference image. StyleAligned demonstrates improved style consistency, but the visual quality of its generated images still falls short compared to our method. StyleShot and CSGO frequently exhibit issues of image duplication and visual artifacts. In contrast, our method produces images that not only exhibit superior style consistency across generated images but also perfectly align with the given prompts.

**User Study.**   We conducted a user study to assess the results of our method compared to other methods. In each question, participants were asked to rank the methods according to the style consistency among their generated images and the alignment of these images with the prompts. In total, our user study involved 40 participants, and Table 2 shows the result. Our method significantly outperforms competing approaches, achieving an average ranking of 1.20, which demonstrates its superior performance.

## 4.3  Ablation Study

**Effect of Shifted Position Embedding (ShiftPE).**   The experimental results in (k) and (m) of Table 3 demonstrate the impact of ShiftPE. When default RoPE used, the $S_{sty}$ and $S_{dino}$ metric rises to nearly 1, revealing a severe content leakage problem. Besides, Figure 8 illustrates that default

| Type | # | PE | $\lambda$ | 0-19 | 19-38 | 38-57 | $S_{text}\uparrow$ | $S_{sty}\uparrow$ | $S_{dino}\uparrow$ |
|---|---|---|---|---|---|---|---|---|---|
| Original | - | - | - | - | - | - | 0.284 | 0.658 | 0.313 |
| Layer | (a) | ShiftPE | 1.10 | ✓ | × | × | 0.283 | 0.669 | 0.323 |
| | (b) | ShiftPE | 1.10 | × | ✓ | × | 0.283 | 0.697 | 0.439 |
| | (c) | ShiftPE | 1.10 | × | × | ✓ | 0.286 | 0.682 | 0.368 |
| | (d) | ShiftPE | 1.10 | ✓ | ✓ | × | 0.274 | 0.697 | 0.420 |
| | (e) | ShiftPE | 1.10 | ✓ | ✓ | ✓ | 0.273 | 0.735 | 0.531 |
| $\lambda$ | (f) | ShiftPE | 0.90 | × | ✓ | ✓ | 0.286 | 0.694 | 0.435 |
| | (g) | ShiftPE | 0.95 | × | ✓ | ✓ | 0.285 | 0.703 | 0.460 |
| | (h) | ShiftPE | 1.00 | × | ✓ | ✓ | 0.286 | 0.711 | 0.485 |
| | (i) | ShiftPE | 1.05 | × | ✓ | ✓ | 0.285 | 0.725 | 0.522 |
| | (j) | ShiftPE | 1.15 | × | ✓ | ✓ | 0.274 | 0.755 | 0.561 |
| w/o ShiftPE | (k) | RoPE | 1.10 | × | ✓ | ✓ | 0.202 | 0.977 | 0.972 |
| w/o SSA | (l) | ShiftPE | 1.10 | × | ✓ | ✓ | 0.281 | 0.733 | 0.525 |
| **Ours** | (m) | ShiftPE | 1.10 | × | ✓ | ✓ | 0.282 | 0.740 | 0.554 |

Table 3: **Ablation study.** The Layer and $\lambda$ denote the ablation study for Layer-Selective Application and scaling factor $\lambda$, respectively. The "w/o ShiftPE" and "w/o SSA" represent the ablation study for the ShiftPE and the SSA module.

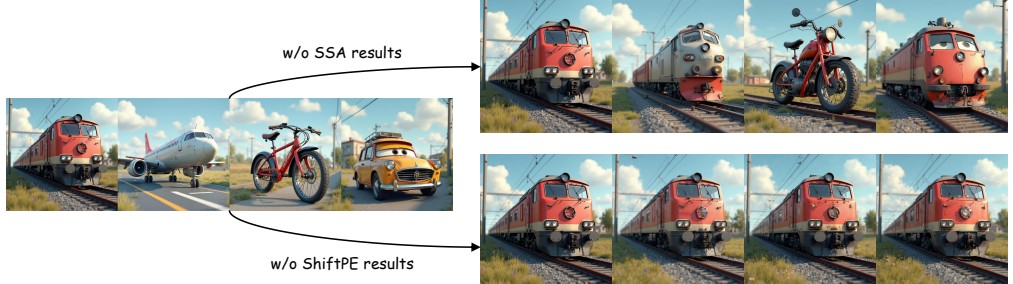

Figure 8: **Visualization of Ablation Study.** The left part presents the result generated by our method. The upper-right image shows the result without the SSA, while the lower-right image displays the result without ShiftPE. The prompt used is "{Train, Airplane, Bicycle, Car} in 3D realism style".

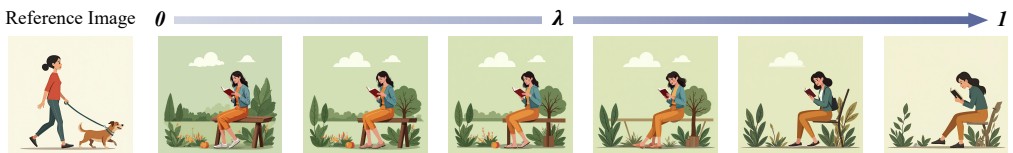

Figure 9: **The impact of scaling factor.** As $\lambda$ increases, the target image progressively aligns with the style of the reference image.

RoPE cause the four generated images to become nearly identical, whereas ShiftedPE maintain style consistency while ensuring alignment with the respective prompts.

**Effect of Selective Shared Attention (SSA).** Table 3 (l) and (m) show results without and with the SSA. Omitting the SSA reduces $S_{sty}$ and $S_{dino}$ but does not improve $S_{text}$. Figure 8 displays outputs under both configurations. Without the SSA, the second image mismatch its textual descriptions (airplane), and style similarity decreases.

**Effect of Layer-Selective Application.** We divided the blocks in Flux.1-dev into three groups: Group 1 contains 19 Double Blocks, Group 2 has the first 19 Single Blocks, and Group 3 includes the remaining 19 Single Blocks. The Layer part in Table 3 shows quantitative results of applying our method in different blocks. Comparing settings (a), (b), and (c) shows that Group 2 blocks contain the richest style information. Comparing settings (b) and (d) reveals that using our method in Group 1 blocks does not improve style consistency and even has a negative effect, which is also confirmed by comparing settings (e) and (m). Comparing settings (d), (e), and (m) demonstrates that applying our

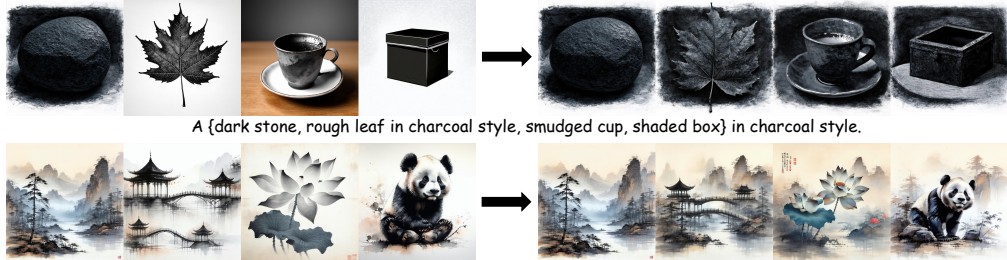

A {dark stone, rough leaf in charcoal style, smudged cup, shaded box} in charcoal style.

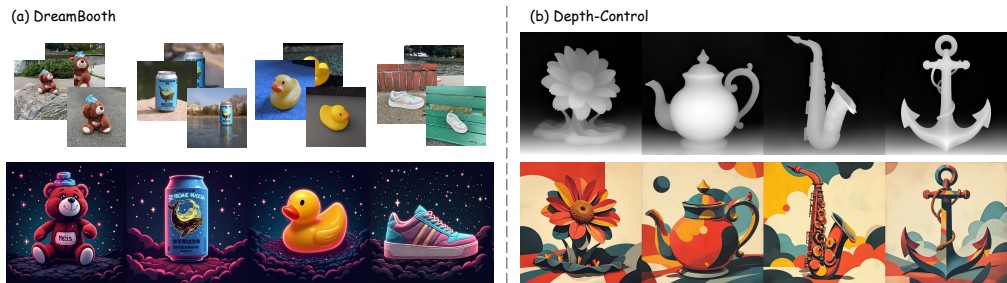

{landscape painting, pavilions and bridges, freehand lotus, panda} in ink wash style.

Figure 10: Generation results of applying AlignedGen to other models. (Top) SD3, (Bottom) SD3.5.

Figure 11: Results from combining AlignedGen with other controllable generation technologies.

method in the last two groups achieves a good balance between image-to-prompt alignment and style consistency across images. Finally, we apply our method exclusively in the Single Blocks (Group 2 and Group 3) to enhance style consistency among generated images.

**Effect of scaling factor.** Table 3 $\lambda$ part presents the quantitative results of our method for various values of $\lambda$. Specifically, as $\lambda$ increases, the style consistency ($S_{sty}$, $S_{dino}$) improves, but this comes at the drop of $S_{text}$. Our experiments indicate that, generally, setting the strength parameter $\lambda$ to 1.1 strikes a favorable balance between style consistency and text alignment. Figure 9 illustrates how the target image progressively aligns with the style of the reference image as $\lambda$ increases.

### 4.4 Generalization and Compatibility

**Architectural Generalization.** To demonstrate our method is not tailored to Flux, we test its performance on other DiT models, including SD3 and SD3.5. As shown in Figure 10, our method seamlessly integrates with these backbones. More results can be found in Appendix C.

**Compatibility with Existing Tools.** A key advantage of our method is its training-free, plug-and-play design. This allows for effortless composition with other generation techniques. As shown in Figure 11, the generated results from combining our approach with other techniques demonstrate the flexibility of our method in practical applications. More cases can be found in Appendix C.

## 5 Conclusion

In this paper, we present AlignedGen, a training-free framework for style-aligned image generation with Diffusion Transformer (DiT). Our approach is built on a critical discovery: the failure of vanilla attention sharing in DiT stems from conflicting position embeddings between reference and target images. We propose Shifted Position Embedding (ShiftPE) to overcome this limitation, unlocking the potential of attention sharing within DiT. Based on this foundation, we further introduce two key enhancements. First, Advanced Attention Sharing (AAS) techniques are specifically designed to refine the attention sharing mechanism for DiT architecture, markedly improving output quality. Second, the feature extraction pipeline broadens the framework's practical applicability, allowing users to provide external image as a style reference. We believe our findings on position embedding and attention sharing could provide insight for future research in style-aligned image generation.

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

## A  Feature Extraction Pipeline

---

**Algorithm 1** Algorithm for Caching Q, K, V from a User-Provided Reference Image

---

**Require:** User-provided reference image $I$, number of inference timesteps $T$
**Ensure:** Cached attention Q, K, V pairs: cached_qkv
1: Initialize a random noise: noise $\sim \mathcal{N}(0, 1)$
2: latent $\leftarrow$ vae_encode($I$)
3: cached_qkv $\leftarrow \{\}$                                      ▷ Initialize an empty dictionary
4: **for** $t \leftarrow T$ **down to** 0 **do**                 ▷ Iterate over $T$ timesteps in reverse
5:      noise_input $\leftarrow \frac{t}{T} \cdot$ noise $+ (1 - \frac{t}{T}) \cdot$ latent     ▷ A standard interpolation forward
6:      $Q_t, K_t, V_t \leftarrow$ DiT(noise_input, $t$, others)             ▷ e.g., prompt embeds
7:      cached_qkv[$t$] $\leftarrow (Q_t, K_t, V_t)$
8: **end for**
9: **return** cached_qkv

---

## B  Additional Compare With Image Editing Methods

Some zero-shot image editing methods can also be applied to style-aligned image generation tasks. We selected two Flux-based editing methods for comparison: RF-Solver [46] and RF-Edit [38]. Figure 12 presents qualitative comparison results between our method and editing methods. Image editing methods often exhibit insufficient editing strength when handling significant image edits such as modifying the main content of the image, leading to persistent retention of original content in the edited results. Table 4 presents the quantitative comparison results between our method and these two methods. As can be observed, our method achieves significantly higher scores on $S_{text}$, $S_{sty}$, and $S_{dino}$ compared to these two baseline methods. It is worth noting that zero-shot editing methods, limited by the instability of editing processes, often require meticulous parameter adjustments to achieve satisfactory results, which limits their practical applicability. In contrast, our method exhibits strong robustness and can typically achieve favorable outcomes in most scenarios without necessitating fine-tuning.

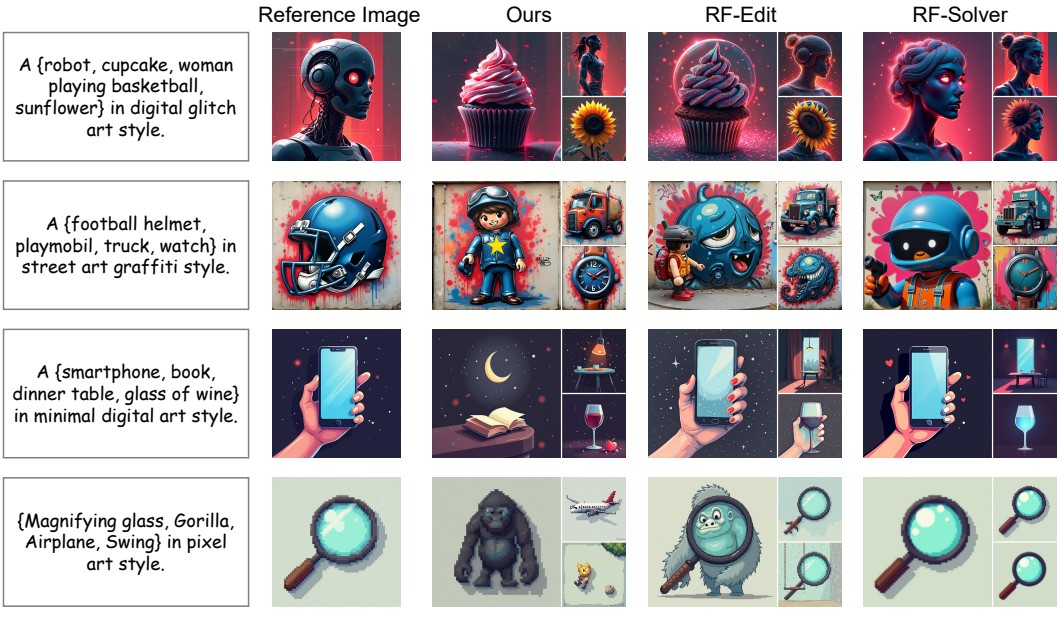

Figure 12: Qualitative comparison between our method and other zero-shot image editing methods.

| Metric | Ours | RF-Solver [46] | RF-Edit [38] |
|---|---|---|---|
| $S_{text} \uparrow$ | **0.282** | 0.278 | 0.280 |
| $S_{sty} \uparrow$ | **0.740** | 0.703 | 0.701 |
| $S_{dino} \uparrow$ | **0.554** | 0.487 | 0.456 |

Table 4: **Quantitative comparison with zero-shot image editing methods.** Best result is marked in **bold**.

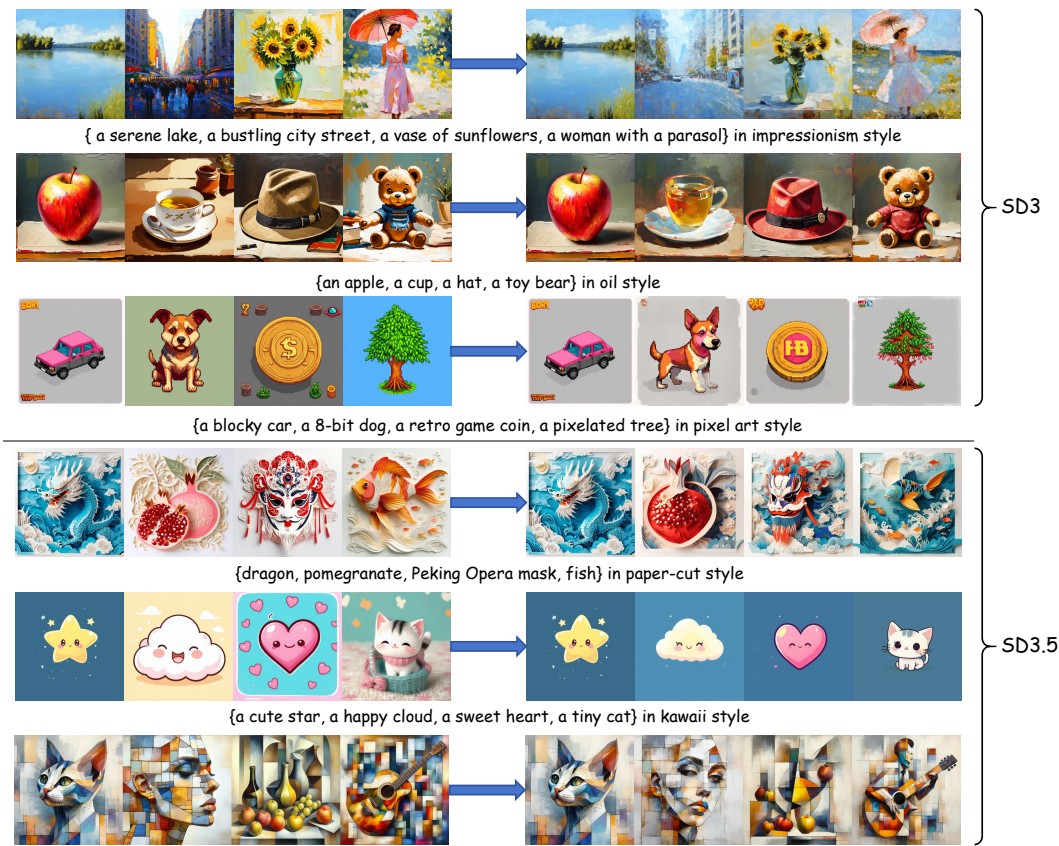

Figure 13: Generation results of applying AlignedGen to other MM-DiT models. The left side presents images generated by the original model, while the right side displays images generated after applying our method.

## C  More Visual Results

Figure 13 presents the generation results of applying AlignedGen to other MM-DiT architecture diffusion models. Figure 14 demonstrates the generation outcomes from combining AlignedGen with Dreambooth. Figure 15 illustrates the generation results achieved by integrating depth control with AlignedGen.

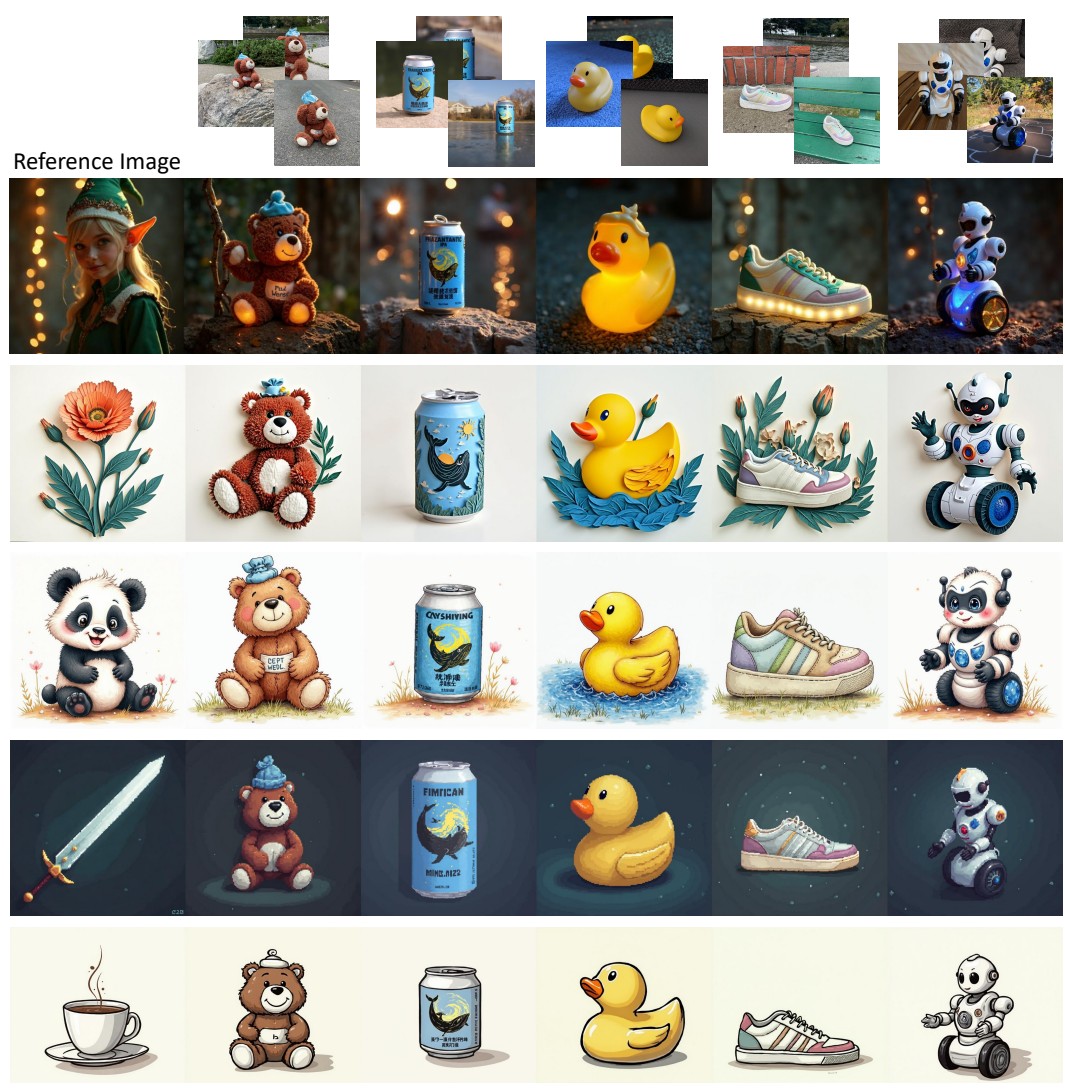

Reference Image

Figure 14: Subject-driven image generation with AlignedGen. Each row shows style aligned image set using the reference image on the left, applied on different personalized diffusion models, fine-tuned over the personalized content on top.

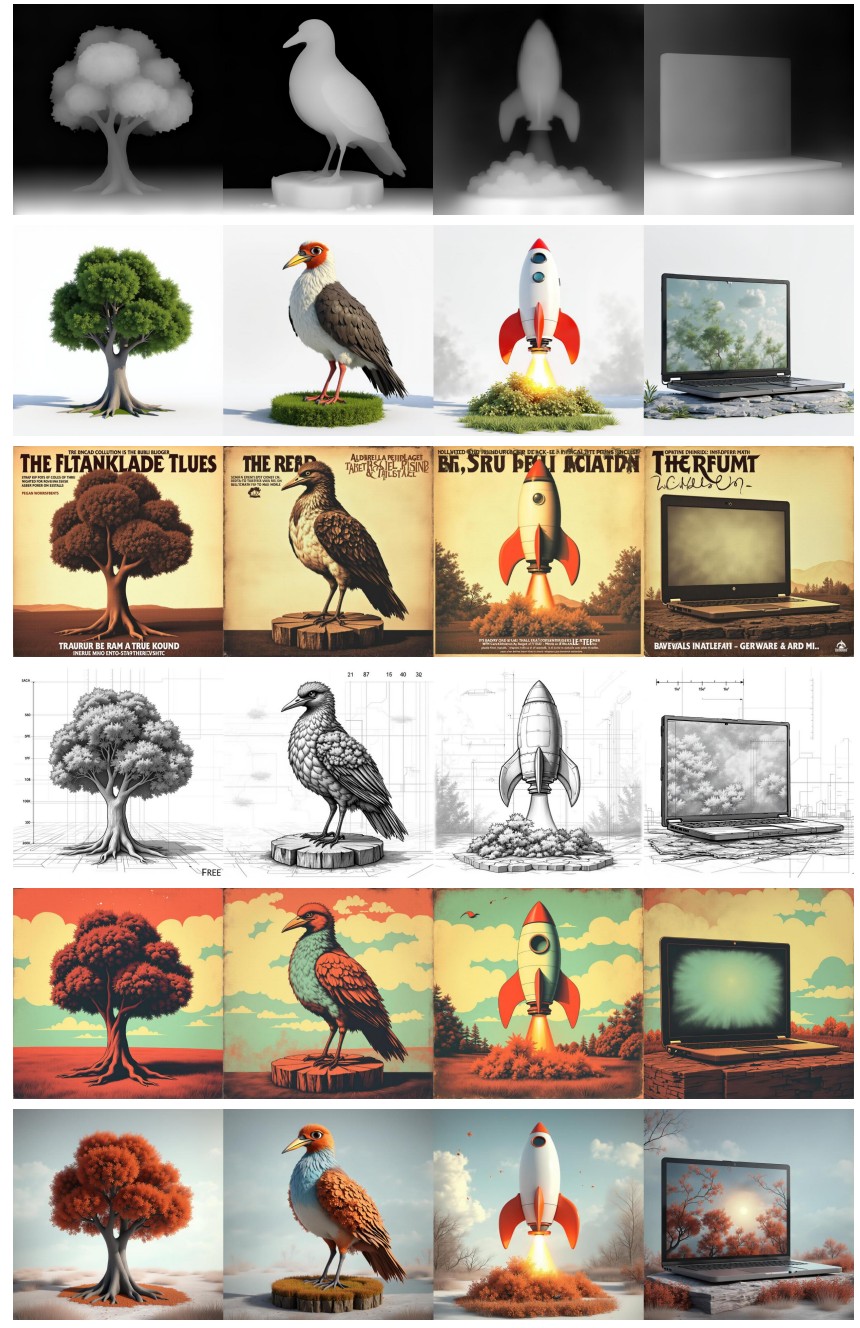

Figure 15: Depth Control with AlignedGen.

