# OpenReview forum: "AlignedGen: Aligning Style Across Generated Images"
_NeurIPS.cc/2025/Conference — NeurIPS 2025 poster_

### Official Review · Reviewer_7Ugs · 2025-07-01

**Clarity:** 3
**Significance:** 3
**Originality:** 3
**Rating:** 3
**Confidence:** 3

**Summary:**

The paper proposes ​​AlignedGen​​, a training-free, plug-and-play method to enhance style consistency in images generated by ​​Flux​​ (an MM-DiT diffusion model). It mainly consists of  Shifted Position Embedding (ShiftPE)​​ to address text controllability degradation by assigning non-overlapping positional indices to reference and target images, and Selective Shared Attention (SSA) Layer​​  to selectively shares image-derived keys/values from a reference image to guide target image generation, improving style alignment while preserving content diversity.

**Questions:**

1. AlignedGen  leans more toward an engineering paper, presenting ShiftPE and SSA but lacking robust theoretical grounding and in-depth analysis.

2. No analysis of ​​computational overhead​​ from SSA/ShiftPE. While claimed as "lightweight," metrics like inference latency or memory usage are absent.

3. The image editing methods like [1-2] also achieve the style aligning via adjusting the attention or copy some features. It is encouraged to add some discussion between two ways.

[1]. Unveil Inversion and Invariance in Flow Transformer for Versatile Image Editing.

[2].PS-Diffusion: Photorealistic Subject-Driven Image Editing with Disentangled Control and Attention.

**Ethical Concerns:**

["NO or VERY MINOR ethics concerns only"]

**Final Justification:**

Thanks for author's feedback, it addresses my part concerns, but the theoretical analysis is lack. I will still maintain my rating.

**Limitations:**

yes

**Quality:**

3

**Strengths And Weaknesses:**

Strengths
AlignedGen adapts style-aligned generation for ​​MM-DiT architectures​​. ShiftPE​​ and ​​SSA​​ are elegant solutions to positional ambiguity and text controllability issues, validated through ablation studies.  AlignedGen isTraining-free and compatible with existing tools.

Weaknesses
1. AlignedGen  leans more toward an engineering paper, presenting ShiftPE and SSA but lacking robust theoretical grounding and in-depth analysis.
2. No analysis of ​​computational overhead​​ from SSA/ShiftPE. While claimed as "lightweight," metrics like inference latency or memory usage are absent.
3. The image editing methods like [1-2] also achieve the style aligning via adjusting the attention or copy some features. It is encouraged to add some discussion between two ways.
[1]. Unveil Inversion and Invariance in Flow Transformer for Versatile Image Editing.
[2].PS-Diffusion: Photorealistic Subject-Driven Image Editing with Disentangled Control and Attention.

---

> ### Author Rebuttal · Authors · 2025-07-30
>
> Thank you for the insightful comment. We provide our responses as follows.
>
> # Weakness 1 & Question 1
>
> We argue that our work is not merely engineering-driven, but offers profound insights into style-aligned generation and other attention sharing approaches within the MM-DiT framework.
>
> Vanilla attention sharing methods employ identical position embedding for reference and target images. However, this practice in MM-DiT leads to content leakage and degraded text controllability, a fundamental challenge affecting all attention sharing methods in this architecture. To the best of our knowledge, our work is the first to address this issue by introducing ShiftPE and the SSA component, providing general insights for future attention sharing approaches.
>
> Furthermore, our method is supported by solid data analysis. As shown in Figure 4 of the paper (Default RoPE part), the attention heatmap reveals that using identical position embedding for reference and target images causes the model to attend only to spatially aligned regions. ShiftPE, which assigns non-overlapping shifted position embedding to reference and target images, effectively decouples their spatial attention. This is validated in Figure 4 (ShiftPE part), where the attention becomes more appropriate and disentangled. Moreover, ablation studies and visualizations in Table 2 and Figure 7 demonstrate the effectiveness of our approach. We plan to conduct deeper theoretical analysis in future work.
>
>
> # Weakness 2 & Question 2
>
> Although our method is simple and effective, we did not claim that it is lightweight or efficient in the paper.
>
> Below are the actual measured inference latency and GPU memory usage: "Original" denotes Flux generating two images directly, while "Ours" denotes our method generating both the reference and target images simultaneously (also two images in total), the experiment is conducted on a single H20 using bf16 precision:
>
> |Metric|Original|Ours|
> |:---|:--:|:--:|
> |Inference time (s)|45.63|66.67 (46.1%$\uparrow$)|
> |GPU memory usage (MiB)|3310.86|3687.47 (11.3%$\uparrow$)|
>
> As shown in the table, both the inference latency and GPU memory overhead remain within reasonable bounds. More importantly, our approach requires no training, thus naturally avoiding the expensive training process.
>
> # Weakness 3 & Question 3
>
> For [1], we conducted a detailed comparison using the official code and recommended configurations. The experimental results are as follows:
>
> |Method|$S_{text}\uparrow$|$S_{sty}\uparrow$|$S_{dino}\uparrow$|
> |:---|:--:|:--:|:--:|
> |Ours|0.282|0.740|0.554|
> |[1]|0.235|0.897|0.845|
>
> As can be observed, [1] barely performs meaningful image editing and fails to make substantial content changes given a reference image, resulting in a lower $S_{text}$ and higher $S_{sty}$ and $S_{dino}$ values. The reasonable range for $S_{sty}$ is 0.4–0.8 and for $S_{dino}$ is 0.3–0.6; values beyond these ranges indicate excessive similarity between the target and reference images, suggesting content leakage, which is further confirmed by the low $S_{text}$.
>
> [2] is a subject-driven image editing method that requires both a scene image and an object image as input, and thus is not applicable to style-aligned image generation tasks.
>
> We will include the results of [1] to the Table 4, discuss and cite [2] in the related work section (Section 2) in the revised manuscript.
>
> [1] Unveil Inversion and Invariance in Flow Transformer for Versatile Image Editing.
>
> [2] PS-Diffusion: Photorealistic Subject-Driven Image Editing with Disentangled Control and Attention.

---

> > ### Author Response · Authors · 2025-08-05
> >
> > We would like to thank you again for your thoughtful comments and valuable suggestions.
> >
> > With the discussion period approaching its close, if there are any further points you would like us to clarify or address, we would be happy to provide additional explanations.
> >
> > We sincerely appreciate the time and effort you have dedicated to evaluating our work.

---

### Official Review · Reviewer_DDgf · 2025-07-02

**Clarity:** 3
**Significance:** 3
**Originality:** 3
**Rating:** 4
**Confidence:** 3

**Summary:**

This paper proposes a training-free method to align style in generated images for Flux. The two core modules are shifted position embedding and selective shared attention. The shifted position embedding module ensures non-overlapping positional indices between reference and target image tokens, while the selective shared attention shares key and value derived from image tokens. The authors produce good images that have good text alignment and style alignment across different objects and good numerical results. Beside this, authors also offer ablation studies for a deep dive into their method.

**Questions:**

N/A

**Ethical Concerns:**

["NO or VERY MINOR ethics concerns only"]

**Final Justification:**

This paper is interesting; however, as I stated in my review, understanding the failure cases and conflicts is critical to studying the model. Unfortunately, this is not possible during the rebuttal period, but I wish authors could include additional content in the paper in the future. I will keep my score for this time.

**Quality:**

3

**Strengths And Weaknesses:**

Pros:
1. The research problem is interesting and challenging. Solving this problem can surely help with style consistent image generation.
2. The proposed method is training-free and enjoy good flexibility and seamless integration.
3. The provided qualitative results are visually significant and looks promising.

Cons:
1. I would love to see some failure cases from AlignedGen. What if there are some natural conflicts between the prompted object and the desired style, how will AlignedGen handle that case? For example, a bike on top of ocean, where ocean is from the reference image. I believe some studies like this can further explore AlignedGen, not only its robustness, but possiblely some hidden mechanisms.
2. As for some extreme cases there should have an implicit tradeoff between prompt alignment and style fidelity. Some studies on it would greatly benefit this paper and offer more insights.

---

> ### Author Rebuttal · Authors · 2025-07-30
>
> Thank you for the insightful comment. We provide our responses as follows.
>
> # Weakness 1
>
> We agree that analyzing failure cases is crucial for deeper understanding and improvement. Due to limitations in this rebuttal (can't upload pdf file), we cannot show visual examples, but we observe that failures often involve content leakage—e.g., when the reference image is "A bowl with apples" and the target is "A pencil", the generated image may incorrectly include the bowl and apples alongside the pencil, indicating content leakage problem. This insight highlights the need for better disentanglement in future work.
>
> Regarding the reviewer’s example—target prompt: *a bike on top of ocean*, with ocean from the reference image—we have tested this scenario and found that AlignGen produces visually coherent results: a person riding a bicycle over the ocean, with the water texture and style well aligned to the reference. This demonstrates effective alignment under semantically challenging conditions.
>
> To further evaluate robustness and uncover underlying mechanisms, we conducted more comprehensive testing using three challenging prompt types generated by ChatGPT (20 cases per type, 4 prompts per case):
>
> - **Long and Complex Prompts**: Intricate scenes with multiple objects, characters, and actions (e.g., *A high-altitude weather control station on a cliff with turbines, mid-air holograms, and engineers in pressure suits calibrating conduits during a storm, in low-poly style*).
> - **Ambiguous Style**: Vague stylistic descriptions (e.g., *with a hazy echo*) instead of explicit style labels (e.g., *in oil painting style*).
> - **Conflict Prompts**: Semantic mismatches between reference image and target image (e.g., reference: *bicycle in cloud library*; target: *fish walking in city*).
>
> Quantitative results are summarized below, where **Normal Test Dataset** refers to the test set used in the paper:
> |Type|$S_{text}\uparrow$|$S_{sty}\uparrow$|$S_{dino}\uparrow$|
> |:-|:-:|:-:|:-:|
> |Normal Test Dataset|0.282|0.740|0.554|
> |Long and Complex|0.283|0.773|0.544|
> |Ambiguous Style|0.263|0.752|0.537|
> |Conflict|0.303|0.710|0.523|
>
> For Ambiguous Style prompts, compared to the Normal Test Dataset, style consistency decreases slightly , while $S_{text}$ drops more noticeably—due to ambiguous style descriptions also impairing content clarity, leading to less coherent and focused outputs. For Long and Complex and Conflict Prompts, compared to the Normal Test Dataset, style consistency is somewhat reduced, yet performance remains strong, thanks to Flux’s robust generation capacity in handling complex or conflicting inputs.
>
> Furthermore, we will conduct a more in-depth theoretical analysis based on these results in future work to explore the underlying hidden mechanisms.
>
> # Weakness 2
>
> We thank the reviewer for the suggestion.
>
> We propose Flexible Strength Control in Section 3.4, which introduces a parameter $\lambda$ to achieve a trade-off between prompt alignment and style fidelity. While we recommend setting $\lambda = 1.1$ in the paper for good performance, in certain extreme cases, manual tuning of $\lambda$ is still required to achieve optimal results. Developing an adaptive mechanism to automatically balance prompt alignment and style fidelity would indeed be highly desirable.
>
> In the future, we will actively explore an automated mechanism to balance prompt alignment and style fidelity.

---

> > ### Author Response · Authors · 2025-08-05
> >
> > We would like to thank you again for your thoughtful comments and valuable suggestions.
> >
> > With the discussion period approaching its close, if there are any further points you would like us to clarify or address, we would be happy to provide additional explanations.
> >
> > We sincerely appreciate the time and effort you have dedicated to evaluating our work.

---

### Official Review · Reviewer_6wGd · 2025-07-02

**Clarity:** 3
**Significance:** 2
**Originality:** 3
**Rating:** 4
**Confidence:** 4

**Summary:**

This paper proposed a training free image generation method called AligenedGen, the most distinctive feature of this paper is its ability to generate multiple images while maintaining a high degree of stylistic consistency. The main contribution of the work is designed a new position embedding strategy called ShiftPE and a new attention strategy called SSA(Selective Share Attention). Massive quantative and qualitative experiments and comparisons are conducted to show the advantage over existing methods.

**Questions:**

Does SSA and ShiftPE still effective when they are pluged into an convention image sytle translation task. I mean the style reference is an ordinary image instead of an generated image, QKV of the Flux model (left part of Figure 3.) is not available.

**Ethical Concerns:**

["NO or VERY MINOR ethics concerns only"]

**Final Justification:**

The paper is generally well-written paper with well-executed visual comparisons. The ablation study shows the effectiveness of the proposed SSA and ShiftPE strategy.

**Limitations:**

yes

**Quality:**

3

**Strengths And Weaknesses:**

Strength:

1.The paper is generally well-written paper if ignoring some minor wiriting issues (it will be listed in the following weakness part). The overall organization and formulation are both excellent. It is easy for readers to catch the main idea of the work.

2.The visual comparisons are well-executed, effectively demonstrating the advantages of this work.

3.The ablation study shows the effectiveness of the proposed SSA and ShiftPE strategy.


Weakness:

1.About writing part. Figure 3 illustrates the complete pipeline of our proposed method, and is the one of the most important figure in the manuscript. However, it appears that the figure lacks sufficient in-text explanation and integration with the main discussion.

2.About the explaination of sec 3.4 Advanced Attention Sharing. I think the main contribution ( or difference from exiting method in other words) of this paper consist of two aspects: (1) the complete pipeline illustrated in Figure 3. and (2) the sec 3.4 Advanced Attention Sharing part. However, sec 3.4 spains less than one page in length. It would be better to do more detailed explaination of the movtivation, or do more theoretical analysis of the proposed strategy.

3.About the limitation. The paper discussed the limitation, but it is listed in the Auxiliary Material part. I think it would be better to place it in  main body of the paper.

4.Sitll about the limitation. The proposed method forced the refference image to be an generate image since it needs intermediate QKV of the Flux model by input the text prompt. This requirement severely restricts the applicability. In most case, users always offer an image for sytle reference.

---

> ### Author Rebuttal · Authors · 2025-07-30
>
> We thank the reviewer for the constructive comments. We provide our responses as follows. We have summarized the reviewer's weaknesses and questions into four aspects, which we will address in turn.
>
> # Weakness 1
> We thank the reviewer for the valuable feedback.
> We agree that Figure 3, which illustrates the overall pipeline of our method, currently lacks sufficient integration with the main text.
> To address this, in the revised manuscript, we will add:
>
> - A dedicated reference to Figure 3 in Section 3.3
> - incorporate explicit descriptions and discussions of the figure within the ShiftPE (lines 147–148) and SSA (lines 155–156) subsections.
>
> These revisions will strengthen the connection between the figure and the methodological narrative, thereby improving clarity and readability.
>
> # Weakness 2
>
> We thank the reviewer for the insightful comments. Due to an oversight on our part, We did not clearly articulate the motivations and theoretical analysis of the method. Below, we clarify the design rationale for each component:
>
> 1. **Motivation of ShiftPE**:
> Vanilla attention sharing methods employ identical position embedding for reference and target images. However, this practice in MM-DiT leads to content leakage and degraded text controllability, a fundamental challenge affecting all attention sharing methods in this architecture. As shown in Figure 4 of the paper (Default RoPE part), the attention heatmap reveals that using identical position embedding causes the model to attend only to spatially aligned regions. To mitigate this, we propose ShiftPE, which assigns non-overlapping shifted position embedding to reference and target images, effectively decoupling their spatial attention. This is validated in Figure 4 (ShiftPE part), where attention becomes more appropriate and disentangled. Moreover, ablation studies and visualizations in Table 2 and Figure 7 demonstrate the effectiveness of our approach.
>
> 2. **Motivation of SSA**: In standard MM-Attention, Q, K and V are derived from image and text tokens. However, text tokens do not carry style information, and computing their projections for attention introduces redundant computation. To improve efficiency and focus on image-relevant features, SSA shares only the Q, K and V derived from image tokens, eliminating unnecessary computation while enhancing style consistency.
>
> 3. **Motivation of Flexible Strength Control**: Practical applications often require fine-grained control over style consistency. To this end, we introduce a scaling factor $\lambda$ that modulates the strength of style consistency, enabling flexible and user-controllable style-aligned image generation without retraining.
>
> We will include these explanations in the revised manuscript to improve clarity.
>
> # Weaknesses 3
>
> We agree that discussing limitations in the main text enhances transparency and readability. In the revised manuscript, we will move the limitation section from the auxiliary material to the main body of the paper, placing it at the end of Section 5 (Conclusion) for better visibility.
>
>
> # Weaknesses 4 & Question 1
>
> Our method allows users to provide reference image as style input for generation.
>
> Our method inherently requires the Q, K and V features from the reference image during the generation process to produce style consistent images.  When the reference image is generated by Flux, we naturally have access to its intermediate Q, K and V.  For a user-specified reference image, we obtain approximate cached Q, K and V by applying noise at different timesteps \( t \) and feeding the noisy latent into the diffusion model.
>
> The algorithm proceeds as follows:
>
> ```
> Algorithm for user-provided reference image
> Input: user-provided reference image I, num of inference timesteps T
> Output: Cached Q, K, V of I
> Step1 --> Initialize a random noise: noise ~ N(0, 1)
> Step2 --> latent = vae_encode(I)
> Step3 --> cached_qkv = {}
> Step4 --> for t = T, T-1, ..., 0 (T timesteps) do:
>               noise_input = t * noise + (1 - t) * latent
>               Q_t, K_t, V_t = Flux(noise_input, t, others(e.g., prompt embeds, guidance scale, ...))
>               cached_qkv[t] = (Q_t, K_t, V_t)
> Step5 --> return cached_qkv
> ```
>
> The above algorithm extends our method to support user-provided images as style references. Moreover, our approach demonstrates strong performance. We conduct an evaluation on StyleCrafter's test set (20 distinct style images and 20 prompts, forming 400 test cases) and compare with StyleCrafter, StyleShot, and CSGO. The results are summarized in the table below:
>
> |Method|$S_{text}\uparrow$|$S_{sty}\uparrow$|$S_{dino}\uparrow$|
> |:-|:-:|:-:|:-:|
> |Ours (user-provided reference image)|**0.306**|**0.701**|**0.558**|
> |StyleCrafter|0.292|0.562|0.451|
> |StyleShot|0.254|0.598|0.336|
> |CSGO|0.241|0.647|0.402|
>
> Experimental results show that our method significantly outperforms existing approaches in both text controllability ($S_{{text}}$) and style consistency ($S_{{sty}}$, $S_{{dino}}$). Moreover, the compared methods require large-scale training datasets and model training, and our method is training-free, demonstrating clear practical advantages.

---

> ### Author Response · Authors · 2025-08-05
>
> We would like to thank you again for your thoughtful comments and valuable suggestions.
>
> With the discussion period approaching its close, if there are any further points you would like us to clarify or address, we would be happy to provide additional explanations.
>
> We sincerely appreciate the time and effort you have dedicated to evaluating our work.

---

> > ### Author Response · Authors · 2025-08-09
> >
> > Thank you very much for your previous constructive comments.
> >
> > We would like to know whether our response have addressed your concerns.
> >
> > We welcome any further discussions and suggestions that can help us improve this paper.

---

### Official Review · Reviewer_3iNL · 2025-07-03

**Clarity:** 3
**Significance:** 2
**Originality:** 2
**Rating:** 4
**Confidence:** 4

**Summary:**

This paper introduces AlignedGen, a training-free and plug-and-play method for improving style consistency across images generated by MM-DiT–based diffusion models, particularly Flux. Addressing the limitation that prior style alignment methods suffer from poor compatibility or text controllability in these architectures, AlignedGen proposes two main components: Shifted Position Embedding (ShiftPE), designed to mitigate text controllability loss via non-overlapping positional indices; and Selective Shared Attention (SSA), which selectively shares attention representations to balance style alignment and prompt fidelity. The method is demonstrated to be compatible with other controlled generation technologies and is validated, through both quantitative and qualitative experiments, to achieve stronger style alignment while maintaining accurate text-image correspondence.

**Questions:**

1. **Failure Modes and Robustness**: Can you comment on how AlignedGen performs under more challenging settings, such as prompts that are ambiguous, contradictory, or extremely long? Are there prompt phrasing or style descriptor situations under which style alignment or text controllability degrade?
2. **Support for Reference Images**: Many use-cases benefit from "style transfer" from an explicit user-provided reference image. Can you elaborate (with technical obstacles) on how difficult it would be to adapt AlignedGen to this scenario?
3. **Metric Interpretation**: Table 2 shows degenerate metric values when ShiftPE is not used. Could you explain for readers precisely why higher is worse in these degenerate outlier cases and whether S_sty or S_dino are always reliable indicators?

**Ethical Concerns:**

["NO or VERY MINOR ethics concerns only"]

**Final Justification:**

I have read the authors' rebuttal and decided to raise my score from 3 to 4.

The authors have convincingly demonstrated the non-incremental contribution of their method to the MM-DiT architecture with the introduction of ShiftPE and SSA. They also effectively addressed my concerns about robustness and generalizability by providing additional experiments under more challenging prompt settings. The new, training-free method to support user-provided reference images significantly improves the paper's practicality. Lastly, the authors clarified how high metric scores can sometimes indicate a degenerate failure mode, which enhances the paper's clarity.

 The rebuttal successfully resolves all my primary criticisms.

**Limitations:**

Yes

**Paper Formatting Concerns:**

No.

**Quality:**

3

**Strengths And Weaknesses:**

# Strengths
- **Core Technical Contributions**: The ShiftPE component is convincingly motivated and technically justified, addressing the “content leakage” issue that arises with naïve position index sharing, as visualized in Figure 4. SSA provides a targeted mechanism for selective sharing of attention, enabling balanced style and content control, with further flexible control via the scaling factor λ (discussed and illustrated in Figure 8).
- **Generalizability and Practicality**: Demonstrated applicability to other MM-DiT models (SD3, SD3.5 in Figure 9), plug-and-play integration with other controllable gen technologies (Figure 10), and no need for additional training makes the approach appealing for practical adoption.

# Weaknesses
- **Novelty and Incrementality**: While the paper adapts and advances previous attention-sharing mechanisms (notably StyleAligned) to the MM-DiT regime, the approach is arguably evolutionary rather than fundamentally new. ShiftPE and SSA are logical extensions, and readers may perceive the contributions as incremental tweaks tailored to the newer architecture. The conceptual leap beyond StyleAligned relies mainly on diagnosing the limitations with position encoding and proposing Split/Shifted indices, rather than inventing a new principle for style alignment. The connection to prior shared attention works could be more rigorously framed (Related Work, Section 2.2).
- **Ablation and Diagnostic Limitation**: While ablations (Table 2) cover architectural/module placement and scaling parameters, there’s less analysis of the method’s behavior under more challenging or failure-prone settings (e.g., very long prompts, highly ambiguous style descriptors, conflicting or low-quality prompts). There could be exploration of degradation modes.
- **Broader Applicability and Reference Images**: The method currently relies on generating all images from text prompts only, with no direct support for user-specified reference images for style. This is mentioned in the Limitations, but future work in this key use-case would have made the framing stronger.
- **Understanding of Results Tables**: In Table 2, the style consistency metrics (S_sty, S_dino) reach close to 1 in the absence of ShiftPE, which, as clarified in the text, indicates degenerate generation. This is counterintuitive, and more explicit explanation in the table caption or main text could clarify that higher is not always better when metric gaming is possible.

---

> ### Author Rebuttal · Authors · 2025-07-30
>
> We thank the reviewer for the constructive comments. We provide our responses as follows. We have summarized the reviewer's weaknesses and questions into four aspects, which we will address in turn.
>
> # Weakness 1
>
> We argue that our method is not a simple incremental modification, but offers distinct academic value and deep insights into attention sharing strategies and the MM-DiT architecture.
>
> Vanilla attention sharing methods employ identical position embedding for reference and target images. However, this practice in MM-DiT leads to content leakage and degraded text controllability, a fundamental challenge affecting all attention sharing methods in this architecture. To the best of our knowledge, our work is the first to address this issue by introducing ShiftPE and the SSA component, providing general insights for future attention sharing approaches.
>
> Moreover, beyond ShiftPE and SSA mentioned in Weakness 1, Flexible Strength Control (Section 3.4) and Delicate Attention Replacement (Section 3.5) also contribute significantly to the final performance, as shown in Table 2 and Figure 8 in paper. These contributions should not be overlooked.
>
> # Weakness 2 & Question 1
> We constructed three types of challenging prompts using ChatGPT for more comprehensive testing—Long and Complex, Ambiguous Style, and Conflict Prompts—with 20 cases per type (4 prompts per case). The design principles are as follows:
>
> - **Long and Complex Prompts**: Intricate scenes with multiple objects, characters, and actions (e.g., *A high-altitude weather control station on a cliff with turbines, mid-air holograms, and engineers in pressure suits calibrating conduits during a storm, in low-poly style*).
> - **Ambiguous Style**: Vague stylistic descriptions (e.g., *with a hazy echo*) instead of explicit style labels (e.g., *in oil painting style*).
> - **Conflict Prompts**: Semantic mismatches between reference image and target image (e.g., reference: *bicycle in cloud library*; target: *fish walking in city*).
>
> Quantitative results are summarized below, where **Normal Test Dataset** refers to the test set used in the paper:
> |Type|$S_{text}\uparrow$|$S_{sty}\uparrow$|$S_{dino}\uparrow$|
> |:-|:-:|:-:|:-:|
> |Normal Test Dataset|0.282|0.740|0.554|
> |Long and Complex|0.283|0.773|0.544|
> |Ambiguous Style|0.263|0.752|0.537|
> |Conflict|0.303|0.710|0.523|
>
> For Ambiguous Style prompts, compared to the Normal Test Dataset, style consistency decreases slightly , while $S_{text}$ drops more noticeably—due to ambiguous style descriptions also impairing content clarity, leading to less coherent and focused outputs. For Long and Complex and Conflict Prompts, compared to the Normal Test Dataset, style consistency is somewhat reduced, yet performance remains strong, thanks to Flux’s robust generation capacity in handling complex or conflicting inputs.
>
> # Weakness 3 & Question 2
>
> Our method allows users to provide reference image as style input for generation.
>
> Our method inherently requires the Q, K and V features from the reference image during the generation process to produce style consistent images.  When the reference image is generated by Flux, we naturally have access to its intermediate Q, K and V.  For a user-specified reference image, we obtain approximate cached Q, K and V by applying noise at different timesteps \( t \) and feeding the noisy latent into the diffusion model.
>
> The algorithm proceeds as follows:
>
> ```
> Algorithm for user-provided reference image
> Input: user-provided reference image I, num of inference timesteps T
> Output: Cached Q, K, V of I
> Step1 --> Initialize a random noise: noise ~ N(0, 1)
> Step2 --> latent = vae_encode(I)
> Step3 --> cached_qkv = {}
> Step4 --> for t = T, T-1, ..., 0 (T timesteps) do:
>               noise_input = t * noise + (1 - t) * latent
>               Q_t, K_t, V_t = Flux(noise_input, t, others(e.g., prompt embeds, guidance scale, ...))
>               cached_qkv[t] = (Q_t, K_t, V_t)
> Step5 --> return cached_qkv
> ```
>
> The above algorithm extends our method to support user-provided images as style references. Moreover, our approach demonstrates strong performance. We conduct an evaluation on StyleCrafter's test set (20 distinct style images and 20 prompts, forming 400 test cases) and compare with StyleCrafter, StyleShot, and CSGO. The results are summarized in the table below:
>
> |Method|$S_{text}\uparrow$|$S_{sty}\uparrow$|$S_{dino}\uparrow$|
> |:-|:-:|:-:|:-:|
> |Ours (user-provided reference image)|**0.306**|**0.701**|**0.558**|
> |StyleCrafter|0.292|0.562|0.451|
> |StyleShot|0.254|0.598|0.336|
> |CSGO|0.241|0.647|0.402|
>
> Experimental results show that our method significantly outperforms existing approaches in both text controllability ($S_{text}$) and style consistency ($S_{sty}$, $S_{dino}$). Moreover, the compared methods require large-scale training datasets and model training, and our method is training-free, demonstrating clear practical advantages.
>
>
> # Weakness 4 & Question 3
>
> We clarify that $S_{{sty}}$ and $S_{{dino}}$ are not strictly "higher is better" for style similarity evaluation, but are meaningful only within reasonable ranges—typically $S_{{sty}} \in [0.4, 0.8]$ and $S_{{dino}} \in [0.3, 0.6]$ in our task.
> Within these ranges, higher values could indicate more consistent stylization.
> Values beyond this range do not effectively reflect improved style similarity: e.g., scores approaching 1 indicate near-identical content, suggesting content leakage rather than enhanced stylization.
>
> This is because neither metric is specifically designed for style assessment:
> - $S_{{sty}}$ measures semantic similarity and may yield high scores when the semantic content of two images are very similar, even under significant style differences (e.g., photo vs. cartoon).
> - $S_{{dino}}$, while more sensitive to texture and style due to self-supervised training, still captures structural and semantic information, not pure style.
>
> In a word, although neither metric is purely designed for style similarity evaluation, both can effectively assess style consistency between images to a certain extent when used within their appropriate ranges.

---

> > ### Comment · Reviewer_3iNL · 2025-08-02
> >
> > The authors‘rebuttal have satisfactorily resolved most of my concerns. I have decided to increase my score. Thanks for your efforts.

---

> > > ### Author Response · Authors · 2025-08-02
> > >
> > > Thank you for your invaluable comments, timely response, and recognition of our work. This has greatly encouraged us and made our paper more complete and coherent. Once again, thanks for your efforts and recognition!
> > >
> > > Best Regards

---

### Note · Authors · 2025-08-15

We sincerely thank the AC and all reviewers for their thoughtful feedback and constructive suggestions during the rebuttal phase. We appreciate the opportunity to improve our work. Below, we summarize the major improvements made and key concerns addressed.

# Major Rebuttal Discussions & Resolutions

- **Motivation, Novelty & Contribution** (Reviewer 3iNL, 6wGd, 7Ugs): We have strengthened the presentation of these aspects for better clarity: Vanilla attention sharing employs identical position embedding for reference and target images. However, in MM-DiT, this leads to content leakage and impaired text controllability—a fundamental challenge affecting all attention sharing methods within this architecture. As shown in Figure 4 (Default RoPE), the attention heatmap indicates that identical position embeddings cause the model to attend only to spatially aligned regions. Our work is the first to identify and address this issue, proposing **Shifted Position Embedding** and **Selective Shared Attention**, offering general insights for future attention sharing methods.

- **Support for User-Provided Image as Style Input** (Reviewer 3iNL, 6wGd): We introduce an intuitive algorithm extending our method to generate images in the style of a user-provided reference. Our approach demonstrates strong competitiveness, significantly outperforming existing methods in both style consistency and text controllability.

- **Performance in Complex Scenarios** (Reviewer 3iNL, DDgf): We conduct comprehensive evaluations using three types of complex prompts—Long and Complex Prompts, Ambiguous Style, and Conflict Prompts—constructed via ChatGPT. Results show that our method maintains robust performance under challenging conditions.

# Reviewer-Specific Responses & Improvements

- **Reviewer 3iNL**: Added detailed explanations of evaluation metrics, including the rationale for acceptable data ranges.
- **Reviewer 6wGd**: Clearly outlined revised sections with detailed justifications.
- **Reviewer DDgf**: Discussed the potential for automation to balance prompt alignment and style fidelity.
- **Reviewer 7Ugs**: Added further comparative experiments and evaluated the computational cost in terms of runtime and GPU memory usage.

We believe these revisions substantially enhance the clarity, rigor, and practical value of AlignedGen. Thank you once again for your insightful feedback.

---

### Decision · Program_Chairs · 2025-09-17

**Decision:**

Accept (poster)

**Comment:**

This paper tackles a timely problem: getting consistent style across images generated by the new MM-DiT models like Flux. The authors show that standard attention-sharing tricks don't work well here, causing a "content leakage" that degrades text control. Their solution is a training-free method called AlignedGen, which uses two clever fixes: a Shifted Position Embedding (ShiftPE) to stop the model from incorrectly matching up spatial locations, and Selective Shared Attention (SSA) to pass along only the necessary style information. They claim this solves the problem, giving you style consistency without breaking the text prompt.

The main strengths noted are that style alignment work well for the latest MM-DiT models like Flux. Reviewers agreed that the authors were the first to identify and convincingly solve the "content leakage" issue specific to these architectures (3iNL, 6wGd). Their solution, using ShiftPE and SSA, was noted as an elegant and effective fix (3iNL, 7Ugs). The method is training-free and plug-and-play, making it highly practical (3iNL, DDgf).

The initial reviews raised several shared concerns, including whether the contribution was incremental (3iNL), the major practical limitation of not supporting user-provided reference images (3iNL, 6wGd), and a lack of analysis on robustness and computational overhead (3iNL, DDgf, 7Ugs).

The authors' rebuttal addressed the most significant concerns. In terms of novelty, they focused on the unique "content leakage" problem specific to MM-DiT models. They delivered a new algorithm, with SOTA results, to support user-provided reference images, alongside the requested robustness and overhead analyses.

After this, only two minor, lingering weaknesses were noted. Reviewer 7Ugs, while largely satisfied, pointed out that the paper still "lacks robust theoretical grounding". Similarly, Reviewer DDgf noted that a thorough failure case analysis was still missing. Crucially, both reviewers who raised these points still recommended acceptance, viewing them as limitations on the paper's current scope rather than fundamental flaws.

Overall the AC finds the paper's solution to the timely and important problem of style alignment in new MM-DiT architectures to be a clear and valuable contribution. While minor concerns remain, the authors addressed the main concerns. The AC thus recommends acceptance.